# Flood Risk Assessment of Buildings Based on Vulnerability Curve: A Case Study in Anji County

**Shuguang Liu** [1,2], **Weiqiang Zheng** [1], **Zhengzheng Zhou** [1,*], **Guihui Zhong** [1], **Yiwei Zhen** [1] and **Zheng Shi** [3]

[1] Department of Hydraulic Engineering, Tongji University, Shanghai 200092, China
[2] Key Laboratory of Yangtze River Water Environment, Ministry of Education, Tongji University, Shanghai 200092, China
[3] Zhejiang Institute of Hydraulics and Estuary, Hangzhou 310020, China
* Correspondence: 19058@tongji.edu.cn

**Abstract:** Following the huge economic losses and building damage caused by yearly flooding in China, increased attention to flood risk management within the urban and suburban areas is required. This paper provides an example of the flood risk management of suburban buildings in Anji County. The temporal and spatial characteristics of inundation in the study area are simulated and analyzed based on a verified coupled hydrodynamic model. The vulnerability curve of local masonry buildings to flood risk is established from the theory of structural static mechanics and the empirical equation of flood load. According to the consequences of the hydrodynamic model and vulnerability curve, a flood risk assessment of suburban buildings is conducted. The results show that severe inundation will occur once the dikes are broken. In the 20-, 50-, and 100-year return periods, there are, respectively, 43, 286 and 553 buildings at extremely high risk, distributed in almost each building region. Over half involved buildings are high risk. Buildings at low-lying lands should worry about the great hydrostatic actions caused by terrible waterlogging. This approach can be popularized in urban, suburban, and rural areas, aimed at frame, masonry and even informal structure. The results can provide a scientific reference for Anji County to reduce the flood loss and enhance the flood resistance.

**Keywords:** flood; risk assessment; vulnerability curve; buildings; Anji County

## 1. Introduction

Flooding is one of the most ruinous natural hazards in the world [1,2]. As a great hindrance to society's sustainable development, urban flooding, especially in dense areas of population, threatens residents' lives and possessions [3]. The spatial inequalities of flood-exposed areas have increased the difficulties to conduct detailed assessments [4,5]. For instance, a flood of a 50-year return period attacked Thailand in 2011, which caused 65 provinces inundated, over 700 people died and a USD 41.2 billion economic loss [6]. Over the period from 2007 to 2016, Busan suffered several devastating flood damages. The property damage accumulated over the 10-year period is nearly USD 150 million, the highest among the major cities in Korea [7]. In China, flooding is also a frequent disaster to which the government is paying much attention. According to the Ministry of Water Resources of the People's Republic of China, from 1999 to 2018, 1181 lives were lost and over 889,300 buildings collapsed yearly due to flood disasters [8]. With the rapid development of urbanization, there are more and more residential settlements concentrated in suburban areas [9]. Suburban areas, which usually perform the functions of both urban districts and rural regions, are vulnerable to extreme natural hazards such as serious flooding [10]. Generally, the flood control capacity of suburban areas approaches that in rural areas, such as low construction quality and poor drainage systems. Once an extreme flood event brought out large-scale building destruction and collapse, terrible loss and



death would become inevitable [11,12]. Unfortunately, changing environmental conditions are driving worsening flood events, with consequences for cities, local communities, and critical infrastructures [13,14]. Therefore, more concentration should be put on the flood risk assessment of buildings.

Damages caused by floods are broadly classified into two categories: tangibles and intangibles [15]. Tangibles damages are those which can be evaluated and calculated quantitatively based on some property, such as economic terms and capacity state. Correspondingly, intangible damages are ones difficult to measure and expressed numerically, such as health-related loss. Tangibles damages can be classified into two types further: direct damages and indirect damages [16]. Direct damages are those caused by physical contact with floodwater, for example, soaking of furniture, the collapse of buildings, and loss of agriculture. Indirect damages usually refer to the impact of interruption of social activities [17,18].

Approaches to assessing flood vulnerability of residential buildings have been available for several decades [19]. However, in recent years, flood risk management has changed significantly and hence the requirements to vulnerability models. Traditional flood damages are normally estimated from the analysis of insurance claims data, historical flood data analysis, or any combination of these approaches. The results from these analyses are primarily expressed as depth-damage functions, as also called "depth-vulnerability curves". In the case of buildings, depth–vulnerability curves represent the average building damage that occurs at different inundation depths. In the UK, the Flood Hazard Research Centre (FHRC) has completed extensive studies estimating UK flood damage. FHRC's work focused on depth–damage curves using slow-rise depth. Their major publications were in the form of manuals [20,21]. In Italy, depth–vulnerability curves were improved based on well-documented data from an extensive damage to assess the costs of future flood events [22]. In Nepal, the vulnerability curve for wattle and daub houses was reported based on field survey and used in flood hazard mapping [23]. Because these studies focus majorly on monetary loss, inundation depth is often chosen as the only parameter in the vulnerability curves. Water level is usually available in historical records and hazard databases. As suggested in a study carried out in Baden-Württemberg, Germany, the maximum water level during the flood event is responsible for the resulting damage [24]. Depending on the research methods, the vulnerabilities of buildings are represented in absolute monetary terms [25], or as relative value [26]. Although vulnerability curves are restrictedly applicable in similar regions, once developed, less time and resources will be required for future events [27]. As long as the fundamental data are adequate enough, these vulnerability curves can consider complex characteristics such as building types and locations. For instance, Huizinga et al. (2017) provided normalized vulnerability curves of different buildings for each continent based on an extensive literature survey across dozens of countries [28]. However, when the flow velocity is fast and the constructions are obviously abraded and destroyed by the great impact force, the assessment of structural safety is more meaningful instead of economic loss. As for influences on buildings, several flood factors including hydrostatic actions (water depth), hydrodynamic actions (flow velocity), erosion actions (flood duration), buoyancy actions, and debris actions are thought to be critical [21]. In suburban and urban areas, dam-break flooding on account of extreme rainstorms is frequent [29], which means flow velocity, in addition to water depth, also plays a major role. Based on empirical data from mainly the Dale Dyke dam failure in Sheffield, Clausen (1989) divided building damages into three states: inundation damage, partial damage, and total damage [30]. Depth and momentum (product of depth and velocity) of floodwater were mainly considered in Clausen's conclusions. In New Orleans, Pistrika et al. (2010) adjusted Clausen's damage criterion by simplifying the variates [31]. Based on the analysis of flood actions and building resistance, Nadal et al. (2010) used statistical simulation methods to propose vulnerability curves. The differences between riverine floods, storm surges, tsunamis, and soil scour were taken into consideration in the report [32]. De Risi et al. (2013) systematically simulated the processes from rainfall to set-

tlement damage and derive analytical vulnerability curves from the simulation results [33]. Custer and Nishijima (2015) defined four states of buildings (no damaging, cracking, partial collapse, and collapse) based on measured and assumed constructional parameters and calculated vulnerabilities of buildings in different forms [34]. Vulnerability curves are widely used in the risk assessment of buildings [35,36]. In flood disasters, because the relationship between buildings and floodwater is conceptualized, the results are usually biased [37]. As mentioned in Mazzorana et al. (2014), the application of vulnerability calculations is limited to the modeling of single buildings due to high computational demands [38].

In China, usually, there are no completed enough flood records. Once a flood disaster occurred, the local government would collect the damage level over the whole area, but not some more detailed information. For instance, the number of influenced residents, inundated buildings, inundated infrastructures, and economic loss all over a county is commonly recorded, but it is difficult to figure out which buildings are inundated or how much the maximum flow velocity around the building is. Consequently, traditional approaches that depend on history databases are inefficient. Additionally, it is suspicious to ignore the influence of flow velocity in assessment, especially when there are great hydrodynamic actions.

The present study focuses on flood risk assessment of suburban buildings in large-scale areas. Legitimately taking hydrostatic actions and hydrodynamic actions into consideration, a quick and cushy assessment is carried out. A local vulnerability curve is presented based on the theory of structural static mechanics and empirical equation of flood load, using observed and hypothetical parameters. In addition, a coupled hydrodynamic model is established to simulate the dike break flooding in different scenarios and to acquire inundation depth and flow velocity nearby the buildings. Flood risk of buildings is assessed on basis of the vulnerability curve and model results. Although the methodology presented is specifically oriented toward masonry structures, it is general for any other structure types. Considering that "vulnerability curve" is an extensively accepted concept in risk assessments [39–41], "curve" is given a more generalized definition in this paper, including conventional curve, surface, and hypersurface.

## 2. Study Area and Data

### 2.1. Anji County

Anji County (ranging from latitude 30°23′ N to 30°53′ N and longitude 119°14′ to 119°53′) is located in the interior of the Yangtze River Delta, in the north of Zhejiang Province, as shown in Figure 1. 60 km away from the urban center of Huzhou City, Anji County is regarded as the transitional zone between the mountain areas of Zhejiang and the agglomeration in Taihu Rim. The total area of Anji County is approximately 1885.71 km$^2$, comprising eight towns. The terrain in Anji County is complex, titling from the southwest to the northeast. Accounting for 11.5% of the areas, mountain land mainly distributes in the southwest and southeast. Hill occupies nearly half of the county, located in the middle. Around 13% of the area is low hill distributed in the northwest, and the remaining 25.5% is plainly located in the middle and northwest. Anji County is located in a subtropical monsoon zone, with an average annual precipitation of 1548.9 mm. Cyclonic storms and convectional rainfall frequently occurs in the flood season (April to October). Those are the main triggers for flood events that effect buildings and human lives.

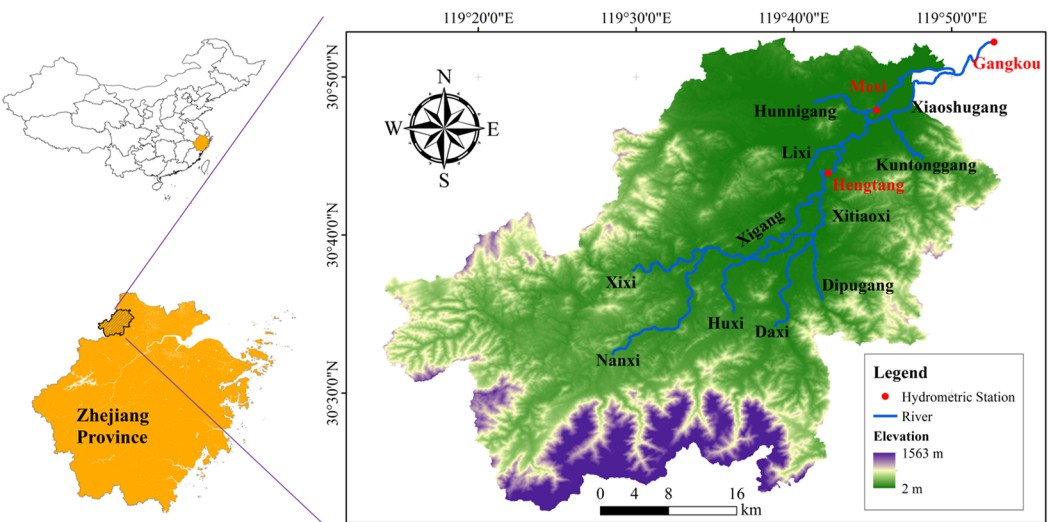

**Figure 1.** Topography and water system in Anji County.

The major river in Anji County is the 111.4 km long river Xitiaoxi (XTX), whose drainage area is approximately 1806 km², as shown in Figure 2. The upstream of XTX is called river Xixi (XX), which is exactly named XTX after converging with river Nanxi (NX) in Dipu Town. XTX flows across Anji County from the southwest to the northeast, successively gathering with rivers Daxi (DX), Huxi (HX), Dipugang (DPG), Xigang (XG), Lixi (LX), Hunnigang (HNG), Xiaoshugang (XSG) and Kuntonggang (KTG) throughout the journey to Taihu Lake. Plum rains and typhoon rainstorms are two major drivers of flood disasters in Anji County. According to the local reports from 1990 to 2016, Anji County suffered from 17 severe floodings. In 1999, strong plum rain attacked Anji County, resulting in over 400,000 residents being influenced, 68,000 buildings damaged, and CNY 59 million lost. In 2013, typhoon Fitow brought a strong rainstorm to Anji County, causing over 200,000 residents to be influenced, 23,000 buildings inundated, and CNY 1.9 billion lost.

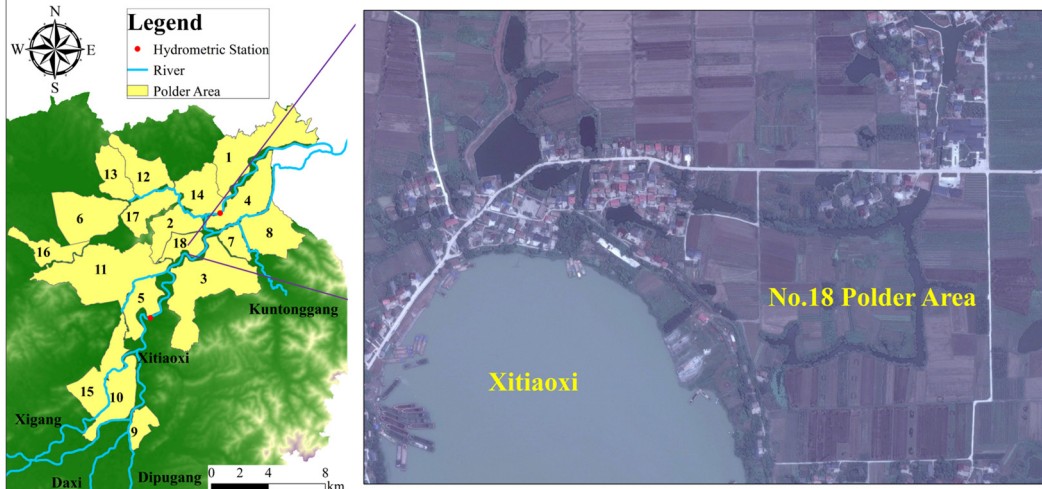

**Figure 2.** Location of polder areas.

### 2.2. Polder Areas

A polder area is a low-lying and easily inundated region that is protected from flooding by surrounding dikes (defending intrusive flooding), sluice gates, and pump stations (controlling internal waterlogging) [42]. Originating at Taihu Basin during the pre-Qin period, polder areas guarantee the development of agriculture and society. However, once the dikes were broken, floodwater would diffuse in the area at a high velocity. For instance, typhoon Fitow (6 to 13 October 2013) resulted in extreme rainstorms that caused

over 20 dike breaks, submerging Tianzihu Town and Meixi Town seriously. According to a comprehensive investigation in 2018, there are 18 polder areas in Anji County (Figure 2). The local dikes can defend against flooding under a 10-year return period, despite several weak fragments. Without loss of generality, the No.18 polder area in Mexi Town is chosen as an example to assess the flood risk of buildings. No.18 polder area is approximately 4.04 km$^2$, located nearby the confluence of XTX and HNG. Most buildings lay neighboring the river, which increases the flood risk.

### 2.3. Data

The cross-section data were acquired in field measurement in 2017, which is thought still applicative enough because riverbeds in plains are thought to be steady. A total of 307 measured cross-sections in 11 rivers were taken into consideration. The land-use type of study area was obtained from satellite remote sensing image maps in 2021, and bathymetry was defined at 5 m × 5 m spatial resolution from Digital Elevation Model in 2021. The boundary conditions consisted of two parts: observed data and designed data. Observed data came from the local hydrometric station during the terms of typhoon Morakot (5 to 11 August 2009) and typhoon Fitow. Designed data in the simulation model, for example, lower boundary conditions in simulation scenarios, came from a local report, *Report of Flood Risk Mapping Project in River Xitiaoxi, Anji County*. Designed hydrologic processes were calculated based on a rainfall–runoff model, in which the results of rainfall frequency analysis relied on the annual maximum method and Pearson-III Distribution.

### 3. Methodology

### 3.1. Hydrodynamic Theories

The water depth and flow velocity are two major factors linked to damage of rural buildings. In order to acquire the values of depth and velocity accurately which play significant roles in risk assessment, a coupled 1D–2D hydrodynamic model was established to simulate the hydraulic conditions of flood based on the DHI MIKE software designed by Danish Hydraulic Institution. The model is considered to be practical and precise. According to the result of field investigation in Anji carried out by the research group, flood disasters are mainly caused by the dike breaks. Consequently, flood caused by dam-break is simulated in this research.

Two major modules, MIKE 11 and MIKE 21, are contained in DHI MIKE where the exchange of water between the inner river and outer area was calculated based on lateral link. The Saint-Venant Equations are numerically solved to simulate and calculate the hydraulic parameters, using an implicit finite difference scheme. These equations are commonly known as a six-point Abbott–Ionescu scheme. The Saint-Venant equations, which show the physical laws of hydrodynamics [43], are denoted by the conservation of mass and momentum equations as shown in Equation (1).

$$\begin{cases} \dfrac{\partial A}{\partial t} + \dfrac{\partial Q}{\partial x} = q \\ \dfrac{\partial Q}{\partial t} + \dfrac{\partial \left( \alpha \frac{Q^2}{A} \right)}{\partial x} + gA\dfrac{\partial h}{\partial x} + \dfrac{gQ|Q|}{C^2 AR} = 0 \end{cases} \tag{1}$$

where $t$ and $x$ represent time and distance; $A$ represents flow area; $Q$ is discharge that passes through $A$; $q$ is later inflow/outflow; $h$ is free surface elevation; $g$ is gravitational acceleration; $C$ is Chezy' resistance coefficient; $R$ is hydraulic radius and $\alpha$ is momentum distribution coefficient.

MIKE 21 model is developed by numerical solution of full, time-dependent, and nonlinear equations of conservation of mass and momentum that are based on depth-averaged Navier–Stokes Equations [44], simple forms of which are shown as Equation (2).

$$\begin{cases} \frac{\partial h}{\partial t} + \frac{\partial(h\bar{u})}{\partial x} + \frac{\partial(h\bar{v})}{\partial y} = hS \\ \frac{\partial(h\bar{u})}{\partial t} + \frac{\partial(h\bar{u}^2)}{\partial x} + \frac{\partial(h\bar{u}\bar{v})}{\partial y} = \bar{v}hf - gh\frac{\partial\eta}{\partial x} - \frac{h}{\rho_0}\frac{\partial p_a}{\partial x} - \frac{gh^2}{2\rho_0}\frac{\partial\rho}{\partial x} \\ \quad + \frac{\tau_{sx} - \tau_{bx}}{\rho_0} - \frac{1}{\rho_0}\left(\frac{\partial s_{xx}}{\partial x} + \frac{\partial s_{xy}}{\partial y}\right) + \frac{\partial(hT_{xx})}{\partial x} + \frac{\partial(hT_{xy})}{\partial y} + hu_sS \\ \frac{\partial(h\bar{v})}{\partial t} + \frac{\partial(h\bar{v}^2)}{\partial y} + \frac{\partial(h\bar{u}\bar{v})}{\partial x} = \bar{u}hf - gh\frac{\partial\eta}{\partial y} - \frac{h}{\rho_0}\frac{\partial p_a}{\partial y} - \frac{gh^2}{2\rho_0}\frac{\partial\rho}{\partial y} \\ \quad + \frac{\tau_{sy} - \tau_{by}}{\rho_0} - \frac{1}{\rho_0}\left(\frac{\partial s_{yy}}{\partial y} + \frac{\partial s_{yx}}{\partial x}\right) + \frac{\partial(hT_{yy})}{\partial y} + \frac{\partial(hT_{xy})}{\partial x} + hv_sS \end{cases} \tag{2}$$

where $t$ represents time; $x$ and $y$ are coordinates in Cartesian coordinates; $h$, $d$, $\eta$ are water depth, surface elevation, time-varying water depth, and $h = d + \eta$; $u$, $v$ are flow velocities in $x$ and $y$ directions; $g$ is gravitational acceleration; $f$ is Coriolis parameter; $\rho$ is density of water; $\rho_0$ is relative density; $p_a$ is atmospheric pressure; $S$ is discharge of point sources; $u_s$, vs. are flow velocities of source item in $x$ and $y$ directions; $s_{xx}$, $s_{xy}$, $s_{yy}$ are components of radiation stress; $\bar{u}$, $\bar{v}$ are mean flow velocities in $x$ and $y$ directions; $T_{ij}$ are components of viscous stress.

MIKE FLOOD model is used to couple the one-dimensional MIKE 11 model and the two-dimensional MIKE 21 model by simulating the momentum transfer between the 1D river network and the 2D surface. In this paper, a lateral coupling method was adopted; water above the river bank is exchanged with the two-dimensional surface model along the flow direction perpendicular to the river, and the exchange flow is approximately calculated by the weir flow formula [45]:

$$Q = WC(H_{us} - H_w)^k\left[1 - \left(\frac{H_{ds} - H_w}{H_{us} - H_w}\right)\right]^{0.385} \tag{3}$$

where $Q$ represents the exchanged discharge; $W$ is the width of connection part; $C$ is the coefficient of weir flow; $k$ is the weir index; $H_{us}$, $H_{ds}$ are the water levels in the upstream and downstream sections of weir; $H_w$ is the elevation at the top of the weir.

### 3.2. Depth–Velocity–Vulnerability Curve

Flow velocity plays significant role in damage to building structures, which is necessary to be taken into consideration together with water depth legitimately in flood risk assessment on suburban buildings [46].

In this paper, vulnerability curve was established based on theories of structural static mechanics and empirical equations of flood impact load, which linked the damage ratio of building structure and water depth, flow velocity. Combining with the results of hydrodynamic model, flood risk assessment could be conducted fast based on the curve. According to the field research, impact and abrasion towards building walls were two major patterns by which flood destroyed the polder area buildings in Anji County, namely bending failure and shear failure. In the following session, unit width wall with one degree of indeterminacy was analyzed on the assumption that flow velocity was uniform distribution vertically [47], as shown in Figure 3.

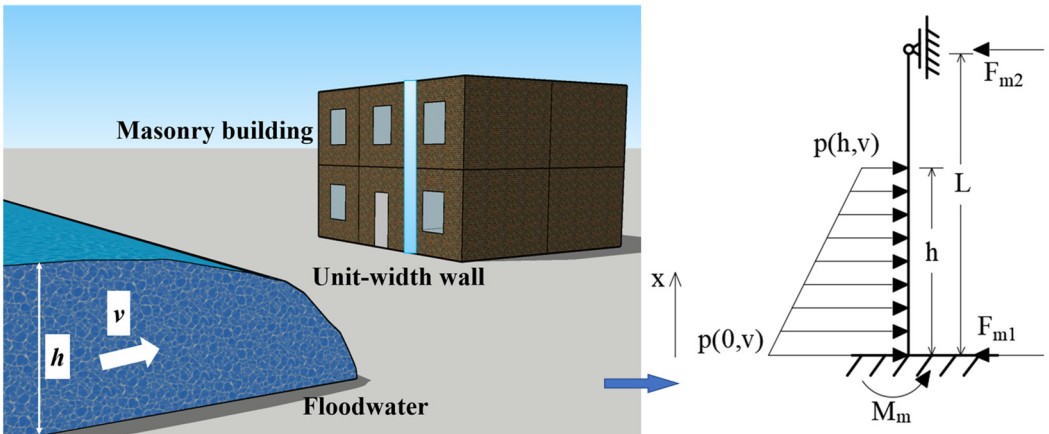

**Figure 3.** Stress condition of wall under flood impacting.

The functional relationships among flood impact load on masonry structures, water depth, and flow velocity were defined by Sun's equation (2011) [48], as shown in Equation (4).

$$p(x,v) = 9637.63 \times (h-x) + 391.07 \times v^2 + 905.91 \times v - 206.63 \tag{4}$$

where $p(x,v)$ in Pa represents impact load; $h$ in m is water depth; $v$ in m·s$^{-1}$ is flow velocity. To be more concise in the following statement, Equation (4) is rewritten as Equation (5):

$$p(x,v) = A(h-x) + f(v) \tag{5}$$

On the assumption that the wall height $L = 3.5$ m, the greatest shearing force ($F_{m1}$ in N) and bending moment ($M_m$ in N·m) can be computed as shown in Equation (6):

$$\begin{cases} F_{m1} = \left| \frac{A}{40L^3}h^5 + \frac{f(v)-AL}{8L^3}h^4 - \frac{f(v)}{2L^2}h^3 + \frac{A}{2}h^2 + f(v)h \right| \\ M_m = \left| \frac{A}{40L^2}h^5 + \frac{f(v)-AL}{8L^2}h^4 - \frac{3f(v)+AL}{6L}h^3 + \frac{A+f(v)}{2}h^2 \right| \end{cases} \tag{6}$$

The ultimate shearing force ($F_c$ in N) and ultimate bending moment ($M_c$ in N·m) can be computed as shown in Equation (7):

$$\begin{cases} F_c = (f_v + \sigma_0) \times \frac{2}{3}t \\ M_c = (f_{tm} + \sigma_0) \times \frac{1}{6}t^2 \end{cases} \tag{7}$$

The density of the wall is $\rho = 1800$ kg·m$^{-3}$, and the thickness is $t = 0.24$ m. The vertical additional stress is $\sigma_0 = \rho g L = 61{,}740$ Pa. According to the designing request of flexural members in *Code for design of masonry structure (GB50003—2011)*, on the assumption that strength class of mortar is M10, ultimate tensile strength of exampled masonry structure is $f_{tm} = 0.27$ Mpa, ultimate shearing strength is $f_v = 0.27$ Mpa.

According to Custer and Nishijima (2015), the security coefficient values 1.2 feasibly [34]. $\xi$ represents the vulnerability of masonry structures, as shown in Equation (8) and Figure 4.

$$\xi = \min\{\max\{F_{m1}/F_c, M_m/M_c\}, 1.2\} \tag{8}$$

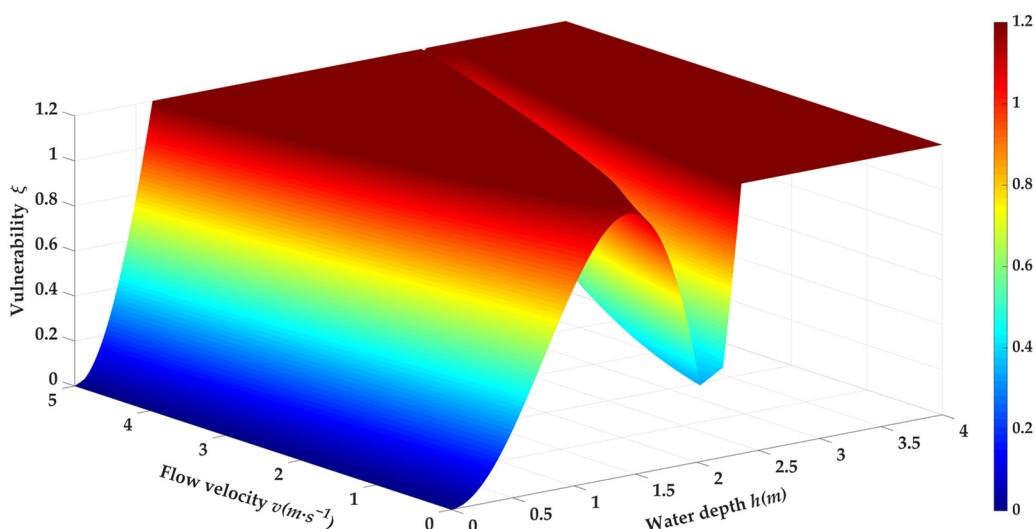

**Figure 4.** Vulnerability curve of masonry buildings.

## 4. Set of Hydrodynamic Model

### 4.1. Set of MIKE 11

Based on the conditions and characteristics of the XTX basin, seventeen rivers were considered in the 1D model. Eleven of the seventeen, namely rivers XX, NX, XTX, DX, HX, DPG, XG, LX, HNG, XSG, and KTG, were involved in the model in detail, using measured cross-section data. The other six, named rivers Gangkougang, Xiangxi, Meiyuanxi, Zixi, Dingshenghe, and Qingshangang, were simplified as point sources in boundary conditions. In the model, the inflow discharge time series of XX, NX, DX, HX, DPG, HNG, and KTG were considered as seven open boundaries. The last eight open boundaries were set at Gangkou Hydrometric Station (GHS), using water level time series. The model was calibrated based on observed water level variation at Hengtang Hydrometric Station (HHS) and Meixi Hydrometric Station (MHS) during typhoon Morakot (2009). As a result, Manning's n was valued within 0.029–0.034 in upstream areas, 0.025–0.03 in downstream areas, and over 0.04 at some meandering zone.

### 4.2. Set of MIKE 21

The 2D model, which contained 53,519 nodes and 106,150 meshes, was divided into unstructured grids. Coordinating the accuracy and efficiency of simulating, meshes in building areas were more intensive (the spatial step was 3 m). Bathymetry was defined at 5 m × 5 m spatial resolution from Digital Elevation Model in 2021 (Figure 1). Horizontal eddy viscosity was computed based on the Smagorinsky formulation, where the constant value was 0.28. Fixed boundaries were adopted, and drying depth, flooding depth, and wetting depth were 0.005 m, 0.05 m, and 0.1 m. Self-adaptive time-step was used in the model between 0.0001 s to 5 s, and flood extent results were saved every 2 min. The floodplain resistance was decided according to the usage of the land, for instance, kept 0.017 in suburban areas, 0.03 in plow areas, 0.04 in bench land, 0.05 in grassland, and 0.18 in forestland. Surface elevation was expressly heightened at the building area so as to simulate the effect of buildings to flood flow (Figure 5). In addition, a pump was considered in the model to simulate the drainage system in the polder, with a capacity of 3.3 m³/s.

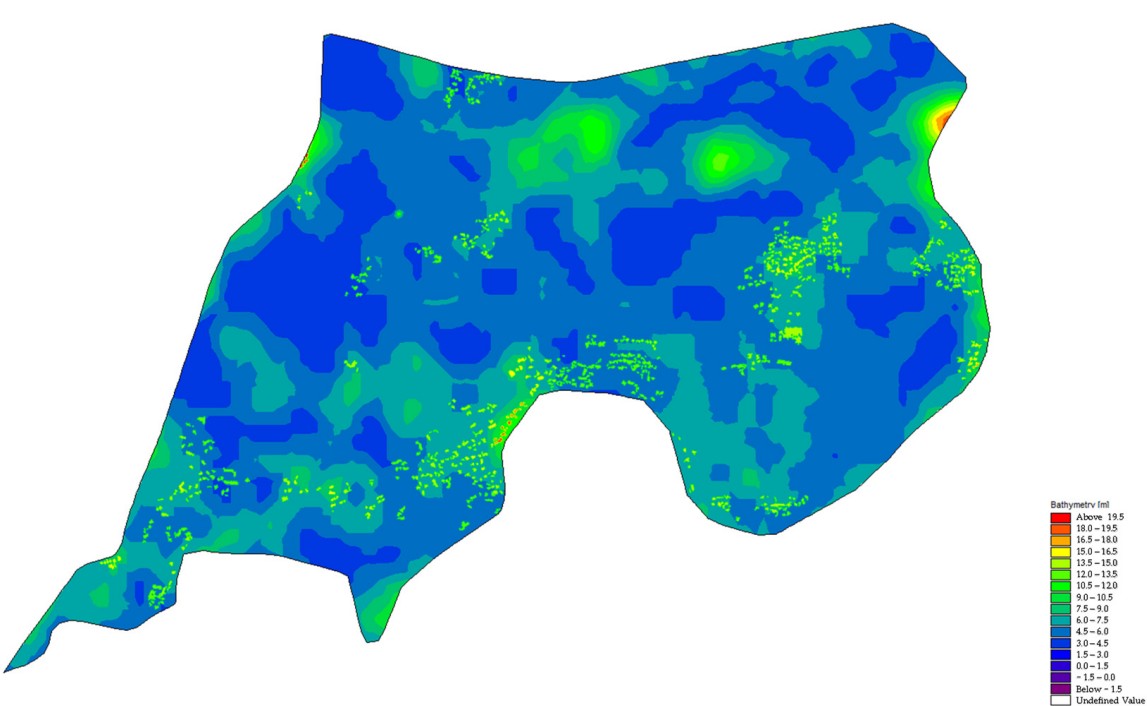

**Figure 5.** Elevation model of study area.

*4.3. Set of MIKE FLOOD*

Dike break was simulated by side structure settings in the coupled model. The river levees in the XTX basin are normative enough to defend against floods below the 10-year return period. Based on statistical data of historical flood hazards and field research, a 20 m long dike break was assumed to occur at Meixi Town (30,750 m chainage of XTX, southeast of No.18 polder area), where the dikes were considered weak. Once the surface elevation reached that in a 10-year return period, a dike break would happen. Situations of 20-, 50-, and 100-year return periods were simulated so that hydraulic factors in the basin were computed. Then, the flood risk of buildings is carried out.

*4.4. Verification of the Coupled Model*

The coupled model was verified on water level and discharge at HHS and MHS for the flood year 2013 during typhoon Fitow. The simulated discharge has shown good agreement with the observed discharge at the measuring locations. A comparison between observed and simulated water levels was found acceptable, as shown in Table 1. As a result, this model performed well in simulating the flooding process of the XTX basin. The coupled model could support the risk assessment of polder area buildings in this study.

**Table 1.** Verification results of coupled model.

| Typhoon | Item | Observed Value | Simulated Value | Error |
|---------|------|----------------|-----------------|-------|
| Morakot | Water level at HHS | 7.61 m | 7.63 m | +0.02 m |
|         | Water level at MHS | 6.70 m | 6.69 m | −0.01 m |
| Fitow   | Water level at HHS | 8.59 m | 8.58 m | −0.01 m |
|         | Water level at MHS | 7.39 m | 7.42 m | +0.03 m |
|         | Discharge at HHS | 1930 m$^3$/s | 1991 m$^3$/s | +61 m$^3$/s |

## 5. Results

### 5.1. Flooding Analysis

The flooding simulation model is used to simulate the situation of the current river channel when it encounters floods with different return periods. The variation of water level at hydrometric stations is significant for flood prevention (Figure 6). XTX basin locates in the transitional area between mountains and plains. With the rise of return periods, the peak values at HHS and MHS grow obviously, and the arrival of peak values advances slightly. The location of HHS is more upriver than MHS, which leads to the larger rangeability and concentration of floodwater at HHS. Considering the ability of local dikes to defend against flooding, a scenario of a 10-year return period is simulated to provide the accordance for dike break settings.

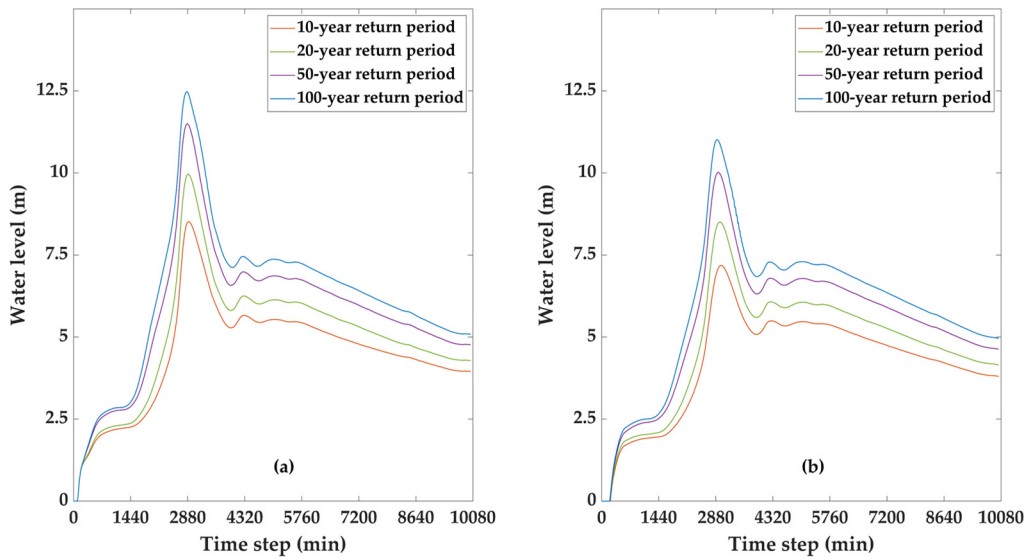

**Figure 6.** Water level series at hydrometric stations in different return periods: (**a**) HHS; (**b**) MHS.

The evolution processes of floodwater in the polder are shown in Figure 7. After flowing into the polder through the dike break, floodwater rushes towards the west rapidly. Meanwhile towards the north, there are two low-lying lands located, respectively, in the middle and western area, resulting in the concentration of floodwater. For a 20-year return period, the dike break occurs at the time of 2725 min. The inundated area increases during the following 10 h, reaching the maximum value of 3.27 km² at nearly 3350 min. It is merely flooding in 20-year return period that generates nearly the whole polder area inundated (Figure 8). For a 50-year return period, the dike break occurs at the time of 2595 min. The inundated area then rises quickly, reaching 3.75 km² at roughly 3250 min. The maximum of water depth reaches over 1.8 m in most low-lying lands (Figure 9). For a 100-year return period, the dike break occurs at the time of 2475 min. After that, the floodwater spreads extremely fast in the polder area, reaching 3.87 km² at the time of 3000 min. Then, the water depth in inundated areas grows gradually, the most severe inundation appears at the time of approximately 3250 min. Severe inundation occurs in the polder, as over two meters of floodwater covers most areas (Figure 10). There are no efficient drainage systems to deal with floodwater inundation in the polder area, except two pumps. Once dike breaks occur, severe waterlogging will spread in the polder area quickly, increasing the flood risk of buildings.

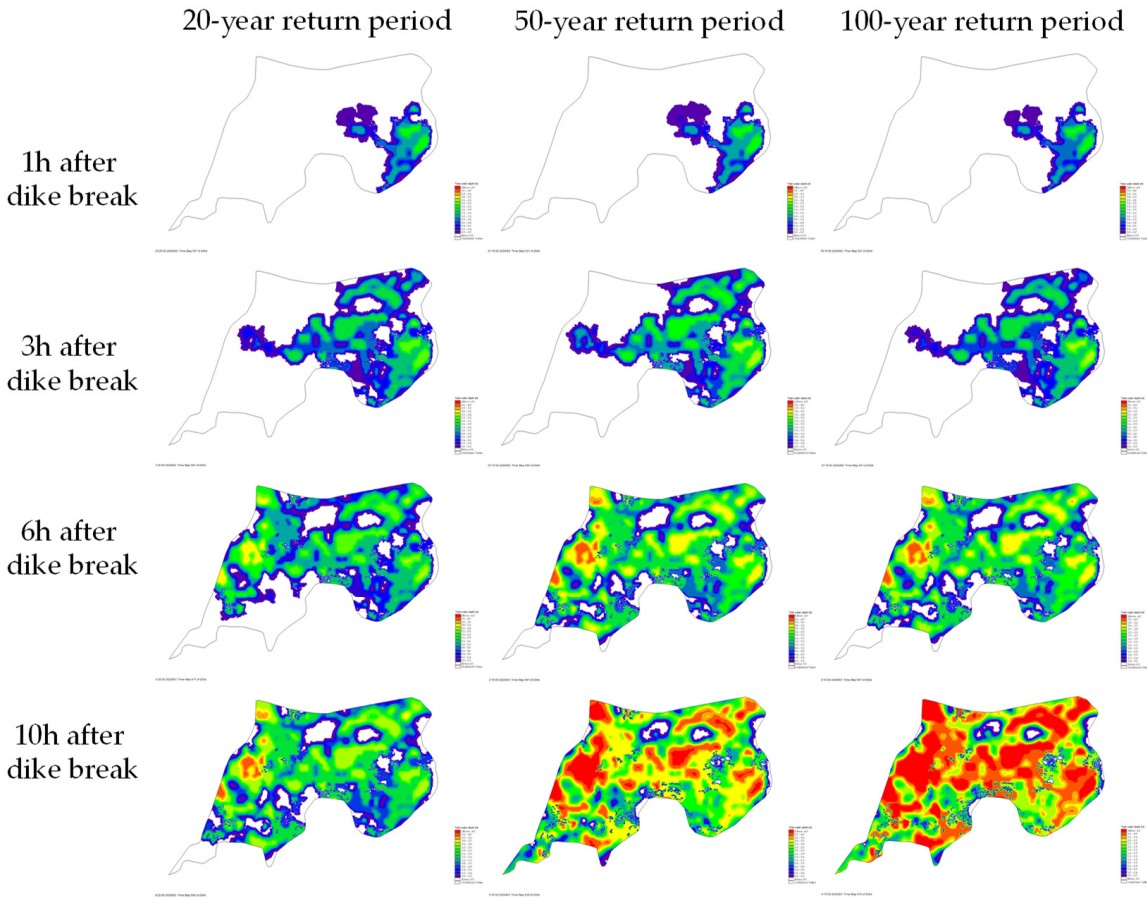

**Figure 7.** Evolution processes of floodwater inNo.18 polder area.

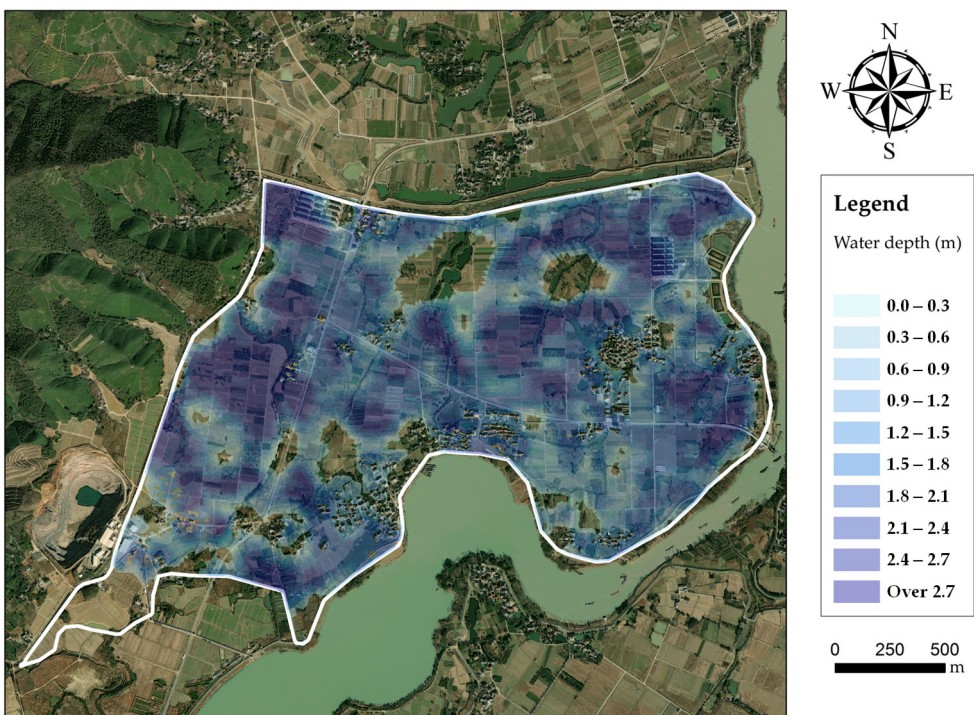

**Figure 8.** Inundated areas in flood scenario of 20-year return period.

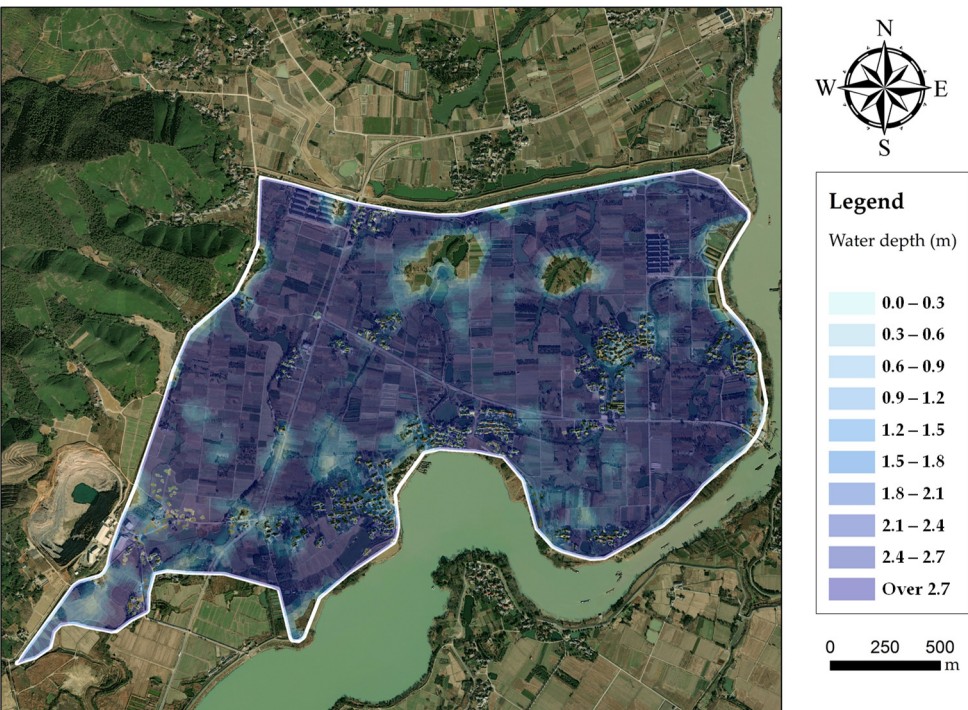

**Figure 9.** Inundated areas in flood scenario of 50-year return period.

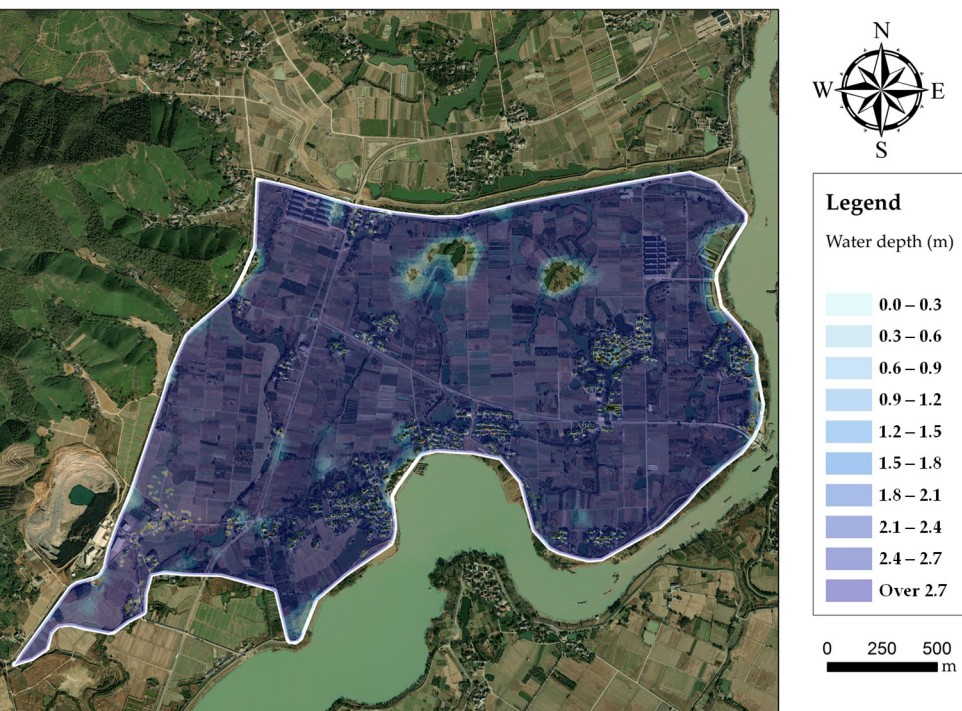

**Figure 10.** Inundated areas in flood scenario of 100-year return period.

*5.2. Flood Risk Assessment of Buildings*

Based on the hydraulic results of simulated flood and vulnerability curves of buildings, flood risk assessment can be conducted fast. Firstly, vulnerability values at every mesh are computed according to Equation (8). Secondly, for each suburban building, weighted mean vulnerability based on surrounding mesh acreage is calculated to represent the vulnerability of this building. Finally, the distributions of buildings under different vulnerabilities in the No.18 polder are presented based on GIS mapping. It is believed that a suburban

building became the ultimate state when its $\xi$ reaches 1.2, consequently, $\xi = 1.2$ is defined as the bound of extremely high risk. In other situations, trisections of the interval [0,1.2], respectively, represent buildings at low, middle, and high risk.

Referring to the Digital Orthophoto Map for 2021, 857 suburban buildings in the study area are analyzed. Amounts of buildings of different vulnerability values in each return period are shown in Figure 11. The abscissa $\xi$ represents the flood vulnerability of buildings, while the ordinate $S$ expresses the total amount of buildings whose vulnerability values are no less than the corresponding $\xi$. For a 20-year return period (Figure 12), 418 buildings are influenced by flooding. There are 43 buildings at extremely high risk, mainly distributed in the southwest, south, and north of the polder area. A total of 216 buildings are at high risk, located dispersedly around the polder. For a 50-year return period (Figure 13), 654 buildings are involved in the floodwater. More than 500 buildings reach high or extremely high risk, mainly located at the periphery of building groups. For a 100-year return period (Figure 14), approximately half of the buildings are at extremely high risk. Almost all buildings that are not on high-lying land or protected by building groups reach high and extremely high risk. According to the results of the flood risk assessment of buildings, the following suggestions are considered to be constructive in preventing and controlling flooding in No.18 polder areas.

1.  Notice the structural safety of dikes. Floodwater can easily diffuse in the polder areas through the dike breaks, due to the lower terrain than the periphery. The assemblage of water neighboring the buildings leads to strong hydrostatic actions while the diffusion of that brings considerable hydrodynamic actions. However, unless the dikes were broken so limited floodwater would invade the polder, several pumps could be adequate.

2.  Take defensive measures around buildings. Bounding walls can reduce the impact between floodwater and buildings and are conducive to draining away floodwater.

3.  Avoid complete isolation caused by floodwater. Actually, strong hydrostatic actions occur only when there is a great water level difference between the exterior and interior. Consequently, appropriately allowing the floodwater to accumulate inside the building is significant in severe flood scenarios.

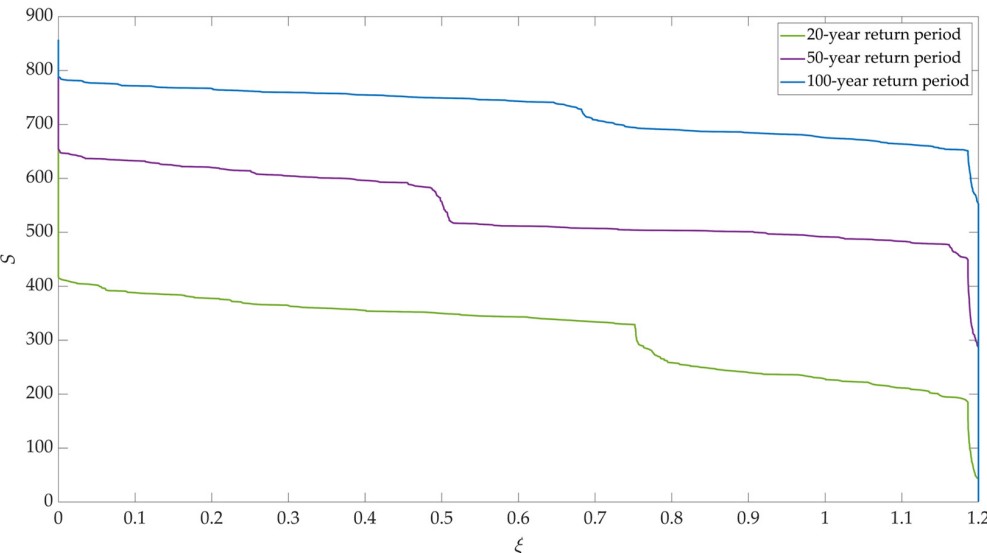

**Figure 11.** Amounts of buildings of different vulnerability values.

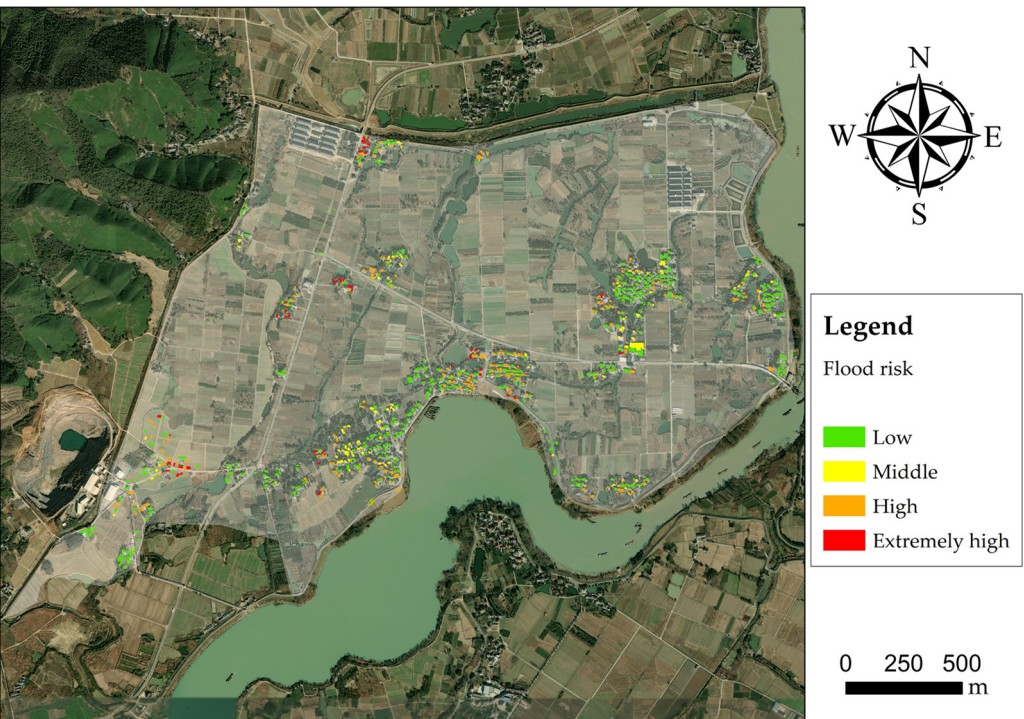

**Figure 12.** Flood risk assessment of buildings in flood scenario of 20-year return period.

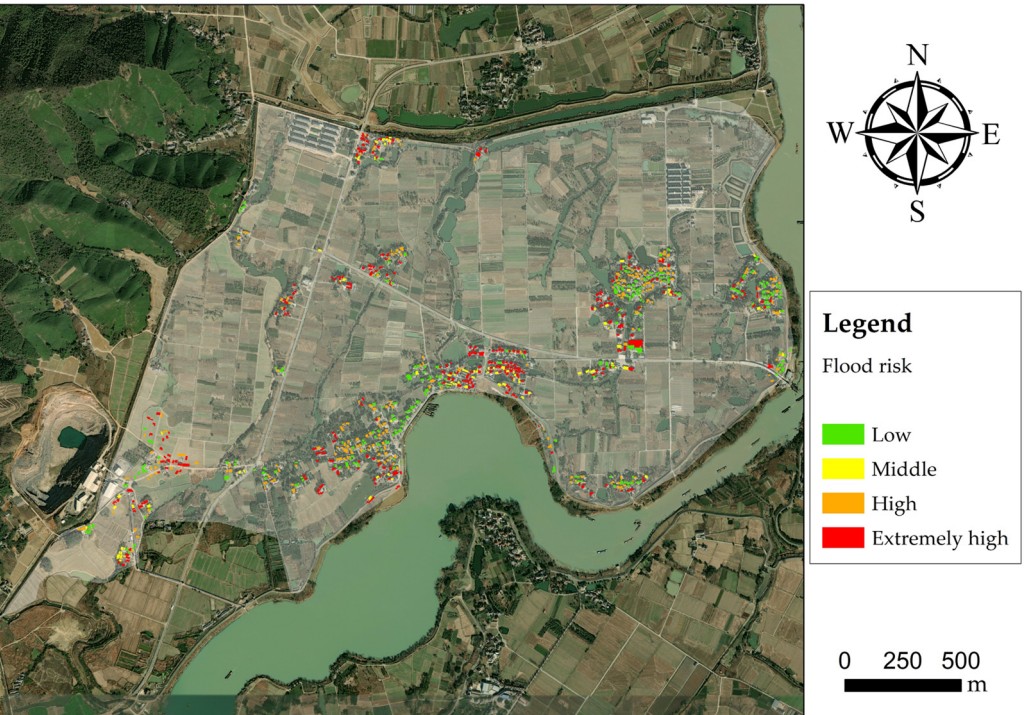

**Figure 13.** Flood risk assessment of buildings in flood scenario of 50-year return period.

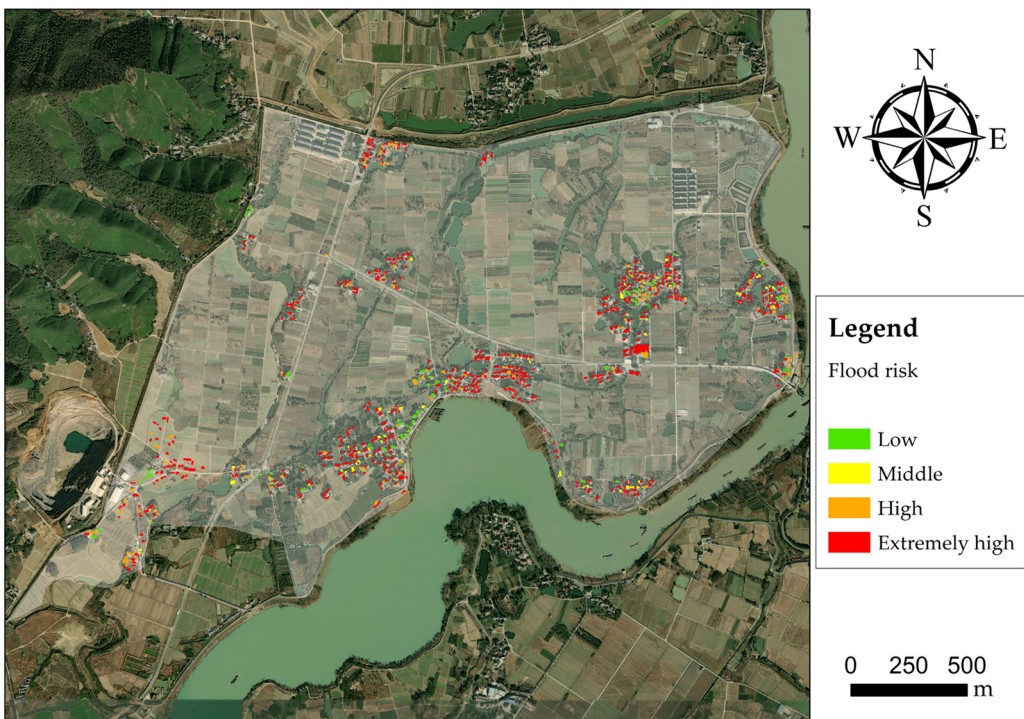

**Figure 14.** Flood risk assessment of buildings in flood scenario of 100-year return period.

## 6. Discussion and Conclusions

### 6.1. Discussion

In this study, hydrostatic actions and hydrodynamic actions are majorly considered in the vulnerability curve. $\xi$ represents the ratio of flood actions and building resistance, which can be regarded as safety redundancy rate to a certain extent. When vulnerability $\xi$ reaches 1.2, the masonry building will transfer into ultimate limit state. When flow velocity $v$ is 0 m/s, the building will be destroyed if water depth $h$ reaches 2.37 m. Additionally, a 0.83 m inundation will be dangerous to the building while $v$ reaches 3 m/s. According to the United States Army Corps of Engineers (USACE), a combination of water depth at 0.83 m and flow velocity at 3 m/s is extremely risky to general buildings [49]. Additionally, Xiao et al. (2013) suggested that masonry buildings will keep at high risk once the flood load reaches 23.7 kPa [50], approximately equal to the result of 24.96 kPa in this study. However, Custer and Nishijima (2015) indicated that the ultimate water depth is over 5 m if there are no hydrodynamic actions while that will be roughly 2.5 m if flow velocity reaches 3 m/s [34]. The differences are attributed to the variant definitions of the ultimate limit state. Custer and Nishijima (2015) used "collapse" which means seriously destroyed, while the strength failure of materials is accepted here. As for decision-makers, targeted measures towards buildings under different vulnerabilities are more requisite, instead of the vulnerability values themselves. Consequently, classifying buildings into different risk rates in accordance with vulnerability values is practically meaningful. Whereas, present studies seldom concentrate on the relationship between the real states of buildings and flood actions at different orders, nor on the quantitative analysis of defensive arrangements. Simple modes are widely used in relative studies, such as the isometric class used in this paper, the qualitative class according to Ettinger et al. (2015) [51], the quantile class mentioned by Moreira et al. (2021) [52], breaks class by Zhen et al. (2022) [53], and so on.

This study provides a new perspective on flood risk assessment of buildings, i.e., relying on scenario simulation. However, due to the lack of historical data, verifying the results is challenging. Although the verified hydrodynamic model and the logical vulnerability curve can guarantee the results, a number of assumptions and limitations have to be pointed out. The expression of building characteristics using assumed and

empirical values is one of the limitations of this study. Moreover, the simplified destruction standard (defined by the ultimate limit state) failed to express the destruction process of buildings. Finally, a single flood impact can be well described by hydrostatic actions and hydrodynamic actions, whereas other long-term influences should be paid much attention to in a systematic assessment. For further studies, the following items should be considered:

1.  The randomness of building parameters. The resistant capacity of buildings relies on a series of random variables, for instance, tensile strength and orientation. The desired method is to ascertain the probability distribution function (PDF) of each factor based on field surveys, physical modeling experiments, and numerical simulations. The vulnerability can be theoretically described as the conditional expectation of destruction probability.

2.  The destruction process is caused by a flood. Flood risk assessment concentrates not only on the ultimate limit state of buildings but also the damage ratio under different situations. Different failure stages correspond to different flood risks. Meanwhile targeted measures are in accordance with the failure stage of buildings. So, the destruction process is the bridge the risk assessment and damage reduction.

3.  The complex actions caused by a flood. Erosion actions, buoyancy actions, and scouring actions play important roles in the long-term flood impact. As for a certain flood, it is the hydrostatic actions and the hydrodynamic actions that bring about the damage to buildings directly. However, other flood actions can gradually weaken the resistant capacity of buildings over a long time scale. That is to say, inundation duration should be a concern more in the assessment of flooding areas.

*6.2. Conclusions*

Flood risk assessment of large-scale buildings is meaningful but arduous. In this paper, a feasible approach is attempted in Anji County. Taking masonry structure as an example, a vulnerability curve is established based on static analysis, considering the joint impact of water depth and flow velocity. The results show that water depth at 2.37 m, or water depth at 0.83 m with flow velocity at 3 m/s is greatly dangerous to buildings. In addition, a coupled hydrodynamic model of the XTX basin is verified to simulate different flood scenarios in Anji County. Due to its location in plain areas, the No.18 polder area will suffer from severe flood threats once dike breaks occur. In the 20-, 50-, and 100-year return periods, approximately 80.9%, 92.8%, and 95.6% of the polder areas get inundated, respectively. Low-lying lands in the middle and west of the polder are waterlogged seriously, with a water depth of over 3 m. Based on the consequences of the vulnerability curve and hydrodynamic model, a flood risk assessment of buildings in the No.18 polder area is conducted. The results show that flooding a high return period can lead to high vulnerability values of buildings. With the return period of flooding increasing from 20 years to 100 years, the number of buildings at extremely high risk rises from 43 to 553. Buildings at high and extremely high risk locate in almost each building region, mainly on account of the flat terrain of the polder. Considering the great influence caused by dike breaks, it is significant to be concerned about the safety of local dikes and it is necessary to take steps to reduce the level difference. Although the vulnerability curve in this paper originated from masonry structures, a similar analysis can be conducted on any type of building after adjusting the parameters and stress analysis.

**Author Contributions:** Conceptualization, S.L., W.Z. and Z.Z.; methodology, S.L., Z.Z. and G.Z.; software, W.Z. and Y.Z.; validation, S.L., Z.Z. and G.Z.; formal analysis, S.L. and W.Z.; investigation, W.Z. and Y.Z.; resources, Z.S.; data curation, S.L., W.Z. and Z.S.; writing—original draft preparation, S.L. and W.Z.; writing—review and editing, S.L. and Z.Z.; visualization, W.Z. and G.Z.; supervision, S.L., Z.Z. and G.Z.; project administration, S.L., Z.Z. and G.Z.; funding acquisition, S.L. All authors have read and agreed to the published version of the manuscript.

**Funding:** This research was funded by the National Key Research and Development Plan during the 13th Five-Year plan period (2018YFD1100401); the National Natural Science Foundation of China (51909191).

**Institutional Review Board Statement:** Not applicable.

**Data Availability Statement:** Not applicable.

**Acknowledgments:** The authors are grateful for the important support given by the helpful comments of all reviewers without which the quality of the paper could not be improved.

**Conflicts of Interest:** The authors declare no conflict of interest. The funders had no role in the design of the study nor in the interpretation of data or the writing of the manuscript.

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
