# Peer review of "Flood Risk Assessment of Buildings Based on Vulnerability Curve: A Case Study in Anji County"

_water, doi:10.3390/w14213572_

Round 1

Reviewer 1 Report

1-      The research gaps should be explained in the introduction.

2-      What is the difference between the current article and other articles?

3-      Can we use methods for other regions of the worlds?

4-      What is your suggestions for the next studies?

5-      How the method can help the decision-makers for controlling flood.

Reviewer 2 Report

The study is interesting as much as it discusses flood risk mitigation and management. The widespread nature of flooding alone makes any study of this nature relevant in any context. I believe that the authors have shown a great mastery of the technique, which has been sufficiently discusses, albeit some aspects of the manuscript require some revision to improve its readability and general research practice and acceptance. 

I am certain that the major weakness in the paper is language and expression. Refer to my correction of abstract in the attached file. The choice of vocabulary is not a good match to academic English. For examples: applicated (line 23, pg.1; lines 414 & 417, pg.16) significative (line 411, pg. 16). Many of the sentences and paragraphs are too long and loose their meaning too early. Some comments are doubtful and unjustified. For example lines 103-104 (pg. 3): where the authors argued: It is worth noting that vulnerability analysis of building clusters rarely appears in the present studies’.

This statement is doubtful and unjustifiable. And when you consider the literature scope of the research, it will be clear literature search and analyses undertaken by the authors are insufficient or just not recent to justify why the study is relevant. There are few recent studies that offer credible insights the authors can use to clarify their rationale for this study: 

1. Ferrito, T., Milosevic, J., & Bento, R. (2016). Seismic vulnerability assessment of a mixed masonry–RC building aggregate by linear and nonlinear analyses. Bulletin of Earthquake Engineering14(8), 2299-2327.

2. Martins, L., & Silva, V. (2021). Development of a fragility and vulnerability model for global seismic risk analyses. Bulletin of Earthquake Engineering19(15), 6719-6745.

3. Andrewwinner, R., & Chandrasekaran, S. S. (2022). Finite Element and Vulnerability Analyses of a Building Failure due to Landslide in Kaithakunda, Kerala, India. Advances in Civil Engineering2022.

4. Papathoma-Köhle, M., Schlögl, M., Dosser, L., Roesch, F., Borga, M., Erlicher, M., ... & Fuchs, S. (2022). Physical vulnerability to dynamic flooding: Vulnerability curves and vulnerability indices. Journal of Hydrology607, 127501.

5. Gautam, D., Adhikari, R., Gautam, S., Pandey, V. P., Thapa, B. R., Lamichhane, S., ... & Rupakhety, R. (2022). Unzipping flood vulnerability and functionality loss: tale of struggle for existence of riparian buildings. Natural Hazards, 1-21.

There is little discussion to this research as much as accuracy assessment was not carried out. The authors need to identify the areas of weakness in this study. 

Reviewer 3 Report

This is a well-written article in the field of flood risk and vulnerability. The outputs are interesting for the journal audience, however some improvements needed based n the reviewer opinion. The main issues are the following:

Line 31. The spatial inequalities of flood exposed areas has raise the concern of scientific community and consist a toolkit for rational urban planning as reported in a recent research (Stefanidis et al., 2022; Chakraborty et al., 2014)

Stefanidis, S., Alexandridis, V., & Theodoridou, T. (2022). Flood Exposure of Residential Areas and Infrastructure in Greece. Hydrology, 9(8), 145.

Chakraborty, J., Collins, T. W., Montgomery, M. C., & Grineski, S. E. (2014). Social and spatial inequities in exposure to flood risk in Miami, Florida. Natural Hazards Review, 15(3), 04014006.

Line 48 Highlight the problem of flooding in fractional urban areas, local communities and critical infrastructures (Mrozik, 2022; Porter et al., 2021)

Mrozik, K. D. (2022). Problems of Local Flooding in Functional Urban Areas in Poland. Water, 14(16), 2453.

Porter, J. R., Shu, E., Amodeo, M., Hsieh, H., Chu, Z., & Freeman, N. (2021). Community Flood Impacts and Infrastructure: Examining National Flood Impacts Using a High Precision Assessment Tool in the United States. Water, 13(21), 3125.

Line 107. Clearly justify the novelty points of the current approach to the best author knowledge

Line 134. Is there any database with flood history data for the study area? It would be useful to add and analyze (spatiotemporal) such data.

Line 213. This discharge is validated with ground true measurements?

The discussion must be further analyzed in depth and discuss the results in a wider context.

Round 2

Reviewer 1 Report

The paper was improved. Thus, I accept it.

Reviewer 3 Report

The authors addressed all the reviewer comments and now the article can be published